# Mineralogy of the Mudeungsan Tuff (Republic of Korea) Using Synchrotron X-ray Powder Diffraction and Rietveld Quantitative Analysis

Donghoon Seoung [1], Pyosang Kim [1], Hyeonsu Kim [1], Hyunseung Lee [2], Min Huh [1,3,4], Hyunwoo Lee [5],* and Yongmoon Lee [2],*

1. Department of Earth and Environmental Sciences, Chonnam National University, Gwangju 61186, Korea; dseoung@jnu.ac.kr (D.S.); kps940@naver.com (P.K.); vbcu132@naver.com (H.K.); minhuh@jnu.ac.kr (M.H.)
2. Department of Geological Sciences, Pusan National University, Busan 46241, Korea; leehs_07@naver.com
3. Mudeungsan UNESCO Geotourism Center, Chonnam National University, Gwangju 61186, Korea
4. Korea Dinosaur Research Center, Chonnam National University, Gwangju 61186, Korea
5. School of Earth and Environmental Sciences, Seoul National University, Seoul 08826, Korea
* Correspondence: lhw615@snu.ac.kr (H.L.); lym1229@pusan.ac.kr (Y.L.)

**Abstract:** Mudeungsan (Mount Mudeung) is an extinct volcano located in the southwestern part of South Korea that was formed in the Late Cretaceous period. This mountain, 1187 m above sea level, is adjacent to Gwangju Metropolitan City, which has a large population (about 1.4 million) and volcanic rocks, including columnar joints, which form various types of outcrops. Although this mountain was listed as a national geopark in 2014 and a UNESCO Global Geopark in 2018, much basic research has yet to be carried out. In particular, there are no mineralogical studies of volcanic rock samples despite the well-preserved variety of volcanic rocks. For this study, X-ray diffraction analysis was conducted using rock samples from Mudeungsan columnar joints known as tuff. We report that the rocks are mostly dacite, mainly composed of quartz, plagioclase, and sanidine through Rietveld quantitative analysis. In particular, α-cristobalite, a crystalline polymorph of silica, appears in the columnar joint rocks, indicating that Mudeungsan experienced an explosive eruption during the formation of the mountain.

**Keywords:** Mudeungsan; X-ray diffraction; Rietveld quantification; cristobalite; volcano

## 1. Introduction

Mudeungsan (1187 m above sea level) is a UNESCO Global Geopark located in the southwest of the Korean Peninsula (Figure 1). Columnar joints are particularly well established around the summit (Figures 1 and 2) and are relatively well preserved (Figure 2). During the Cretaceous period of the Korean Peninsula and East Asia, the Izanagi plate was subducted into the Eurasian plate, and the Korean Peninsula, as a continental arc, had numerous volcanic activities. Cretaceous volcanic rocks are abundant in the southeastern and southwestern regions of the Korean Peninsula [1]. In this regard, many columnar joints formed in the Cretaceous period have been reported in the southwestern part of the Korean Peninsula, showing traces of volcanic eruptions [2]. It is believed that columnar joints of Mudeungsan were formed by lava flows [2,3]. Kim et al. (2002) reported a welded structure and proposed the Mudeungsan columnar joint to be welded tuff. In addition, Jung et al. (2014) suggested that Cheonwangbong and Anyangsan in the Mudeungsan region were formed from calc-alkaline magma in the continental subduction zone [4]. Lim et al. (2015) reported formation ages in the range from 84.73 to 87.72 Ma through zircon U-Pb dating of Mudeungsan tuff samples [5]. Although the age of Mudeungsan and the origin of magma have been studied, no research has been conducted on the minerals in the welded tuff constituting the columnar joints. Therefore, in this study, X-ray diffraction

(XRD) analysis was performed on the Mudeungsan tuff samples for the first time in order to report the constituent minerals through Rietveld quantification.

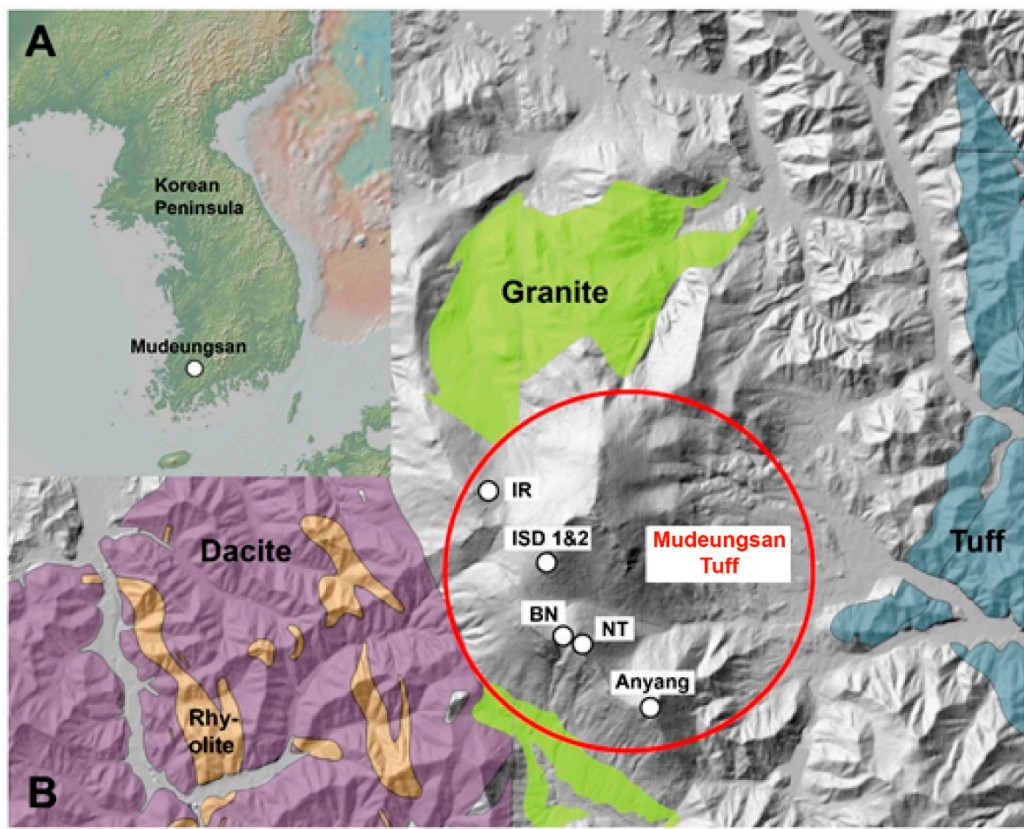

**Figure 1.** (**A**) Location of Mudeungsan in the southwestern Korean Peninsula. (**B**) Geological map of Mudeungsan (geological information was obtained from Lim et al. (2015)). Locations of the Mudeungsan tuff samples are shown as white circles. The area marked with a red circle was reported to be Mudeungsan tuff by Lim et al. (2015).

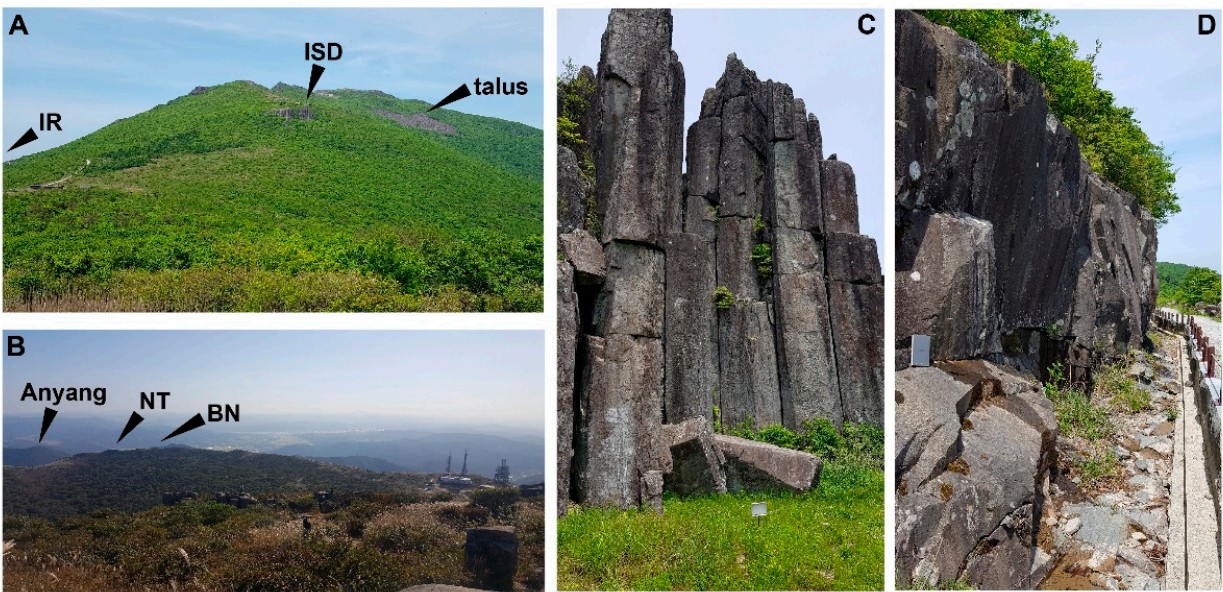

**Figure 2.** (**A**) Ipseokdae rock (ISD) and talus that was formed by the collapse of columnar joints are shown. Ice rock (IR) is located beyond the scope of this photo. (**B**) Baekma ridge (BN), Nakta peak (NT), and Anyang peak. (Anyang). BN is the ridge of the mountain. (**C**) Close-up view of ISD. (**D**) View of IR. This was observed along the trail.

## 2. Materials and Methods

Whole-rock samples of the Mudeungsan tuff were processed into powder, and $SiO_2$, $Al_2O_3$, $TiO_2$, $Fe_2O_3$, $K_2O$, $Na_2O$, CaO, MgO, MnO, and $P_2O_5$ abundances were measured using X-ray fluorescence spectrometer (SHIMADZU XRF-1700) at the at the National Center for Interuniversity Research Facilities (NCIRF) at Seoul National University (SNU), Republic of Korea. Analytical error was within 0.1–1% according to rock-based standards, and loss on ignition (LOI) values were calculated by weight before and after igniting powder samples at 950 °C. Measured major element compositions are shown in Table 1.

**Table 1.** Major element compositions of the Mudeungsan tuff samples (in wt.%).

| | $SiO_2$ | $Al_2O_3$ | $TiO_2$ | $Fe_2O_3$ | MgO | CaO | $Na_2O$ | $K_2O$ | MnO | $P_2O_5$ | LOI |
|---|---|---|---|---|---|---|---|---|---|---|---|
| **ISD-1** | 64.64 | 16.16 | 0.58 | 4.61 | 1.55 | 4.11 | 3.79 | 3.17 | 0.09 | 0.19 | 1.08 |
| **ISD-2** | 64.53 | 15.91 | 0.56 | 4.41 | 1.51 | 4.09 | 3.83 | 3.22 | 0.09 | 0.19 | 1.61 |
| **IR-1** | 64.73 | 15.75 | 0.57 | 4.52 | 1.60 | 4.21 | 3.83 | 3.11 | 0.10 | 0.19 | 1.34 |
| **IR-2** | 64.53 | 15.56 | 0.58 | 4.59 | 1.58 | 4.00 | 3.78 | 3.31 | 0.10 | 0.19 | 1.75 |
| **BN-1** | 65.09 | 16.07 | 0.58 | 4.44 | 1.44 | 3.86 | 3.75 | 3.34 | 0.08 | 0.19 | 1.06 |
| **BN-2** | 64.83 | 15.57 | 0.57 | 4.45 | 1.45 | 3.84 | 3.77 | 3.23 | 0.09 | 0.19 | 1.98 |
| **NT-1** | 64.24 | 15.93 | 0.59 | 4.60 | 1.56 | 4.08 | 3.79 | 3.20 | 0.09 | 0.19 | 1.70 |
| **Anyang** | 64.55 | 15.54 | 0.59 | 4.52 | 1.48 | 3.67 | 3.57 | 3.34 | 0.09 | 0.19 | 2.39 |
| **Average** | 64.64 | 15.81 | 0.58 | 4.52 | 1.52 | 3.98 | 3.76 | 3.24 | 0.09 | 0.19 | 1.61 |

The high-resolution synchrotron XRD analysis of the selected rock samples was measured at the 9B beamline of Pohang Accelerator Laboratory (PAL). Incident X-rays were generated by deceleration of the electron bundle as they passed through the bending magnet. Its average spectral luminous intensity is about 10,000 times higher than that of X-rays extracted from laboratory sources. The incident beam was vertically bonded by a mirror and monochromatized to a wavelength of 1.5419(1) or 1.5225(1) Å using a double-crystal Si(111) monochromator. The detector arm of the vertical scan diffractometer is composed of 6 sets of soller slits, flat Ge(111) crystal analyzers, anti-scatter baffles, and scintillation detectors, with each set separated by 20 degrees. The soller slits and crystal analyzers play a role in removing noise signals such as inelasticity or air scattering. A good peak-to-background is achieved by very bright synchrotron X-ray beams and instrumental devices, which can detect very weak signals (i.e., very low quantity of phases) in diffraction patterns. Each specimen of ca. 0.2 g powder was prepared using the flat-plate side-loading method to avoid the preferred orientation, and the sample rotated normally with respect to the surface during measurement to increase sampling statistics. The step scan was performed at room temperature from 7° to 121° in 2 θ with 0.005° increments and 2° overlaps to the next detector bank. Samples were exposed for 4 s at each stage to increase the degree of normalization of the diffracted beam intensity.

Qualitative and quantitative phase analysis of minerals in rock samples was derived from Rietveld refinement using the Match!3 program (Crystal Impact, Bonn, Germany) [6,7]. The background curve was manually and graphically determined. A quadratic derivative method was used to retrieve Bragg's peaks [8]. The pseudo-Voigt function proposed by Thomson et al. was employed for the profile shape function of the diffracted pattern [9]. 'PDF-2' and 'crystal open database' were used as mineral databases. In order to determine the integrated peak intensity for quantitative phase analysis, the full width of the half maximum (FWHM) of each peak was calculated by the profile-fitting method. The weighted residual factor (R-factor) of all analyses was measured to see that fit was achieved between experimental X-ray diffraction data and crystallographic models. The R-factor can be mathematically defined by the following equation:

$$R = \frac{\sum \|F_{\mathrm{obs}}\| - \|F_{\mathrm{calc}}\|}{\sum |F_{\mathrm{obs}}|}$$

where $F$ is a so-called structural coefficient related to the reflective strength, and the sum means all X-ray reflections of the measured and calculated parts. Analytical errors calculated from the R-factor and included in parentheses (Table 2) were less than ca. 6% when analyzing all samples, excluding the ISD-2 (11.7%) sample. In ISD-2, the R-factor was comparatively higher than the other values due to the unidentified peaks. The results of qualitative and quantitative phase analysis are summarized in Table 2 and Supplementary Information.

**Table 2.** Quantitative phase analysis results using Rietveld refinement (in%).

|  | $_wR_p$ | Quartz | Plagioclase | Sanidine | $\alpha$-Cristobalite | Chlorite | Biotite | Zeolite | Sum |
|---|---|---|---|---|---|---|---|---|---|
| **ISD-1** | 5.8(1) | 26.8(1) | 62.7(1) | 9.2(1) | 0.6(1) |  | 0.6(1) |  | 99.9(1) |
| **ISD-2** | 11.7(1) | 29.0(1) | 58.2(1) | 9.9(1) | 0.7(1) | 0.9(1) | 1.1(1) | 0.2(1) | 100.0(1) |
| **IR-1** | 5.7(1) | 25.1(1) | 56.9(1) | 15.8(1) | 1.1(1) | 0.5(1) | 0.5(1) |  | 99.9(1) |
| **IR-2** | 5.4(1) | 33.6(1) | 49.3(1) | 14.2(1) | 2.7(1) |  | 0.1(1) |  | 99.9(1) |
| **BN-1** | 5.7(1) | 22.3(1) | 61.5(1) | 11.6(1) | 2.6(1) | 2.0(1) |  |  | 100(1) |
| **BN-2** | 4.3(1) | 32.5(1) | 47.7(1) | 19.3(1) | 0.4(1) | 0.1(1) |  |  | 100(1) |
| **NT-1** | 4.3(1) | 21.9(1) | 64.4(1) | 13.0(1) | 0.6(1) |  | 0.1(1) |  | 100(1) |
| **Anyang** | 4.8(1) | 29.9(1) | 55.7(1) | 11.7(1) | 0.1(1) | 1.1(1) | 0.4(1) | 1.1(1) | 100(1) |

## 3. Results

Based on the results of the X-ray fluorescence spectroscopy (XRF), the amounts of the main components follow an order of $SiO_2 > Al_2O_3 > Fe_2O_3 > CaO > Na_2O > K_2O > LOI > MgO > TiO_2 > P_2O_5 > MnO$ (Table 1). Since the standard deviation of each major component is less than 0.42 wt.% (Table 1), the columnar joints of Mudeungsan have a somewhat homogeneous chemical composition. According to the total alkali–silica diagram, all tuff rocks are displayed in the area of dacite (Figure 3) [10]. According to the geological map of this area (Figure 1), dacite was reported to the southwest of the Mudeungsan area, and the study area can also be expanded with the same rock distribution.

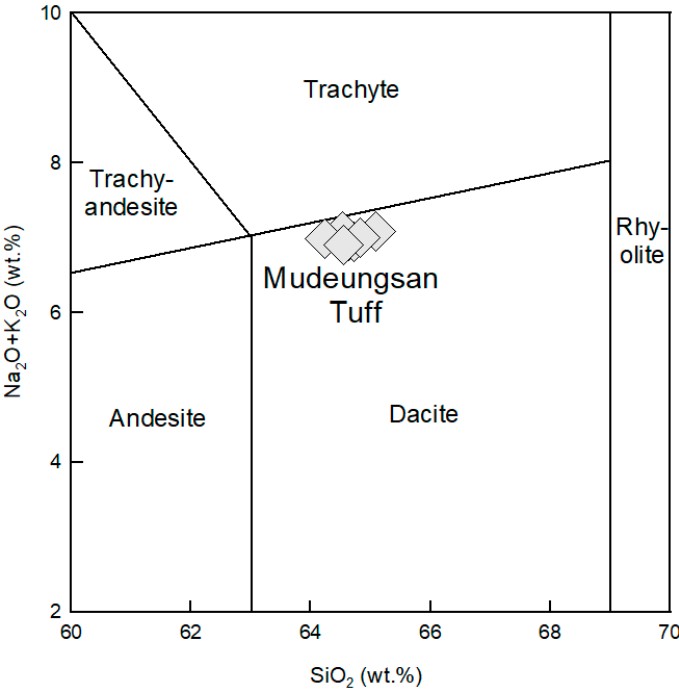

**Figure 3.** Total alkali–silica (TAS) diagram [10] using chemical analysis results (Table 1).

The stacked synchrotron X-ray diffraction patterns and phase identification results are shown in Figure 4. In most cases, identified peaks belonging to quartz and plagioclase are predominantly observed. Sanidine, cristobalite, biotite, and altered minerals such as chlo-

rite and zeolite coincide with minor phases. The quantitative phase analysis results of the selected minerals using the Rietveld refinement method are summarized in Table 2. For the ISD-2 sample, the $d_{(001)}$ reflection of biotite (at ca. 0.6 Å$^{-1}$) appears in the preferred orientation, in contrast to the homogeneous results of the XRF analysis. Therefore, this reflection is not considered in the Rietveld analysis stage to avoid overestimation of the amount of biotite. The samples are mainly composed of quartz (21.9 to 33.6%) and plagioclase (47.7 to 64.4%). Sanidine are observed as a major component, ranging from 9.2 to 19.3%. Mudeungsan is characterized by the presence of cristobalite (0.1 to 2.7%). Small amounts of chlorite, biotite, and zeolite are found. It is expected that biotite can be converted into chlorite by changing two layers of biotite to one layer of chlorite through hydrothermal reaction. Zeolite is known to occur in volcanic ash and rocks with alkaline water at an appropriate temperature. These minerals therefore may indicate that the Mudeungsan tuff is slightly altered.

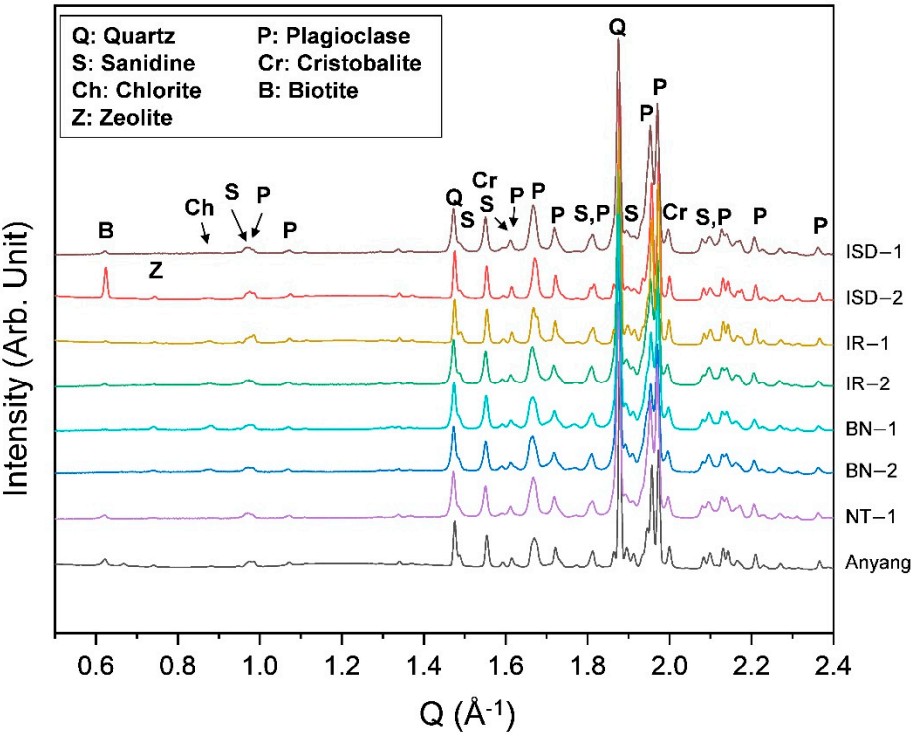

**Figure 4.** High-resolution synchrotron X-ray powder diffraction patterns of the Mudeungsan tuff samples.

## 4. Discussion

The Mudeungsan tuff provides evidence of volcanic activity by producing high-temSSperature minerals such as sanidine and cristobalite. In this section, we focus on cristobalite, which was first reported in the Mudeungsan area. Cristobalite, a crystalline polymorph at a high-temperature and low-pressure in the SiO$_2$ system, is a metastable silica phase in the surface environment. Nevertheless, it has been widely reported in extrusive rocks [11] and particularly in volcanic ash formed in dome-forming eruptions [12,13].

Our synchrotron X-ray diffraction results show that the Mudeungsan tuff contains mainly α-cristobalite belonging to the P4$_1$2$_1$2 space group in the tetragonal system. α-cristobalite appears when cooled to approximately 240 °C or less, accompanied by a displacive transition from cubic β-cristobalite to tetragonal α-cristobalite [11,13–15]. In volcanic rocks that have recently erupted, β-cristobalite can be observed, and both α and β types may also appear together [13]. However, no β-cristobalite is shown in the Mudeungsan samples, which may be due to the age of these rocks (>80 Ma). Thus, these samples were preserved for a long time after cooling and the β-α cristobalite transition

could be completed during the process. However, it is worth considering that such silica phases did not completely convert to quartz, a stable phase under ambient conditions.

The α-cristobalite content of the Mudeungsan tuff samples (0.4 to 2.7%) is significantly lower compared to those reported in modern active volcanoes such as the Soufrière Hills (up to 11%) in Monserrat and Cordón Caulle (up to 23%) in Chile [11,15]. It can be inferred that the columnar joints have progressed from α-cristobalite to quartz to some extent. The time when cristobalite was converted to quartz at room temperature was proved to be less than 105 years [16]. However, it has been suggested that it could take up to tens of millions of years to cool slowly from cristobalite to quartz [17]. Some argue that hydrothermal fluids produce cristobalite through recrystallization of vapor in vesicles in relation to the explosive dome-forming eruptions [15,18]. Volcanic cristobalite, in particular, has been proposed to be produced in the process of vapor-phase crystallization or devitrification in volcanic ash [12]. Sanidine and cristobalite were also reported in the Cretaceous Paraná Magmatic Province (southern Brazil), indicating that they were mainly produced by the devitrification of dacites and rhyolites [19].

Mudeungsan lacked discussion about its long-standing volcanic activities despite the distribution of volcanic rocks. Therefore, we would like to suggest that the existence of the first reported cristobalite in the columnar joint samples does not only report mineralogical information but also that Mudeungsan experienced an explosive eruption at that time.

## 5. Conclusions

Mudeungsan, located in the southwestern part of South Korea, was formed in the late Cretaceous period. At the top of the mountain, the dacitic tuff consists of columnar joints, which allowed Mudeungsan to be registered as a national geopark and a UNESCO Global Geopark. In this study, a synchrotron XRD analysis was conducted for the Mudeungsan columnar joint samples, and quantitative information on major minerals was provided through Rietveld refinement. Quartz and plagioclase are major minerals, and characteristically high-temperature minerals such as sanidine and α-cristobalite were identified in all samples. As no β-cristobalite is found in the Mudeungsan tuff, it is inferred that β-cristobalite has been converted to alpha-type mostly during cooling below 240 °C. Cristobalite is formed from the devitrification or recrystallization of vapor into vesicles, which is mainly known to be associated with explosive volcanic eruptions. Therefore, we propose that Mudeungsan columnar joints were formed by tuff through an explosive eruption at the time.

**Supplementary Materials:** The following are available online at https://www.mdpi.com/article/10.3390/app112210796/s1. Figure S1: a. Graphical result of phase identification and quantitative analysis of the sample ISD-1; b. Graphical result of phase identification and quantitative analysis of the sample ISD-2; c. Graphical result of phase identification and quantitative analysis of the sample IR-1; d. Graphical result of phase identification and quantitative analysis of the sample IR-2; e. Graphical result of phase identification and quantitative analysis of the sample BN-1; f. Graphical result of phase identification and quantitative analysis of the sample BN-2; g. Graphical result of phase identification and quantitative analysis of the sample NT-1; h. Graphical result of phase identification and quantitative analysis of the sample Anyang.

**Author Contributions:** Conceptualization, D.S., H.L. (Hyunwoo Lee) and Y.L.; methodology, P.K., H.K., and H.L. (Hyunseung Lee); software, P.K., H.K., and H.L. (Hyunseung Lee); formal analysis, H.L. (Hyunseung Lee); investigation, P.K., and H.K.; resources, M.H., and D.S.; writing—original draft preparation, H.L. (Hyunwoo Lee), and Y.L.; writing—review and editing, H.L. (Hyunwoo Lee), and Y.L.; funding acquisition, M.H., and D.S. All authors have read and agreed to the published version of the manuscript.

**Funding:** This study is supported by Gwangju Metropolitan City, Republic of Korea (20190801F50-01). Also, we were financially sponsored by National Research Foundation of the Ministry of Science and ICT of Korean Government, grant number NRF-2019K1A3A7A0910157413.

**Data Availability Statement:** The data presented in this study are available on request from the corresponding author.

**Acknowledgments:** We thank the staff of Mudeungsan National Park for their cooperation.

**Conflicts of Interest:** The authors declare no conflict of interest.

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
