# Peer review of "Mineralogy of the Mudeungsan Tuff (Republic of Korea) Using Synchrotron X-ray Powder Diffraction and Rietveld Quantitative Analysis"

_applsci, doi:10.3390/app112210796_

Round 1
Reviewer 1 Report
The manuscript Mineralogy of the Mudeungsan tuff (Republic of Korea) using synchrotron X-ray diffraction and Rietveld quantitative phase analysis contain interesting qualitative study of pyroclastic rocks. It is well organized and written, although it suffer of not really good written abstract, and not satisfactory discussion. In my opinion methodology and the results should be supported by microscopic study. Description of investigated rock is lacking and the term “tuff” should be explained. Abstract and discussion should be rewritten. The manuscript can be publish after major revision.
Lines 53-54 it is not clear, what kind of rocks are marked on gray? Tuffs? Granites? Sample description say that they are tuffs but is not clear on the Figure 1
Lines 59-62 Please, explain the abbreviations in the caption of Figure 2;
Line 63 Methodology – synchrotron XRD method should be extended of description of the analytical errors particularly for substances of quantity below 1% , R parameter should be explain.
Line 64 How many samples have been analyzed?
Line 109 there is not possibly to show altered minerals without microscopic studies. Zeolites could be the primary one, as hydrothermal phases , the chlorites commonly forms from biotite thus biotite is altered to chlorite
Line 112 in the Table 1 the contents of sanidine is completely different
Line 115 not clear which minerals has suffered alteration, mineralogical description and petrography of the studied here samples is lacking.
Line 120. Low cristobalite can be pedogenic origin, it is observed as a detrital mineral in soils formed on the tuffs and occurs commonly as detrital grains in soils that formed on pyroclastic volcanic materials (Cronin et al., 1996, Sommer et al., 2006, Gutierrez – Castorena, Effland 2010) and this possibility should be taken to account into discussion as the sample were taken from the surface.
Line 128 synchotron X-Ray diffraction instead of XRD
Lines 147 - 150 what about devitrification in studied tuffs ?
Line 154 – the explosive eruption have not been proved in this work.
Author Response
Point 1: Lines 53-54 it is not clear, what kind of rocks are marked on gray? Tuffs? Granites? Sample description say that they are tuffs but is not clear on the Figure 1
Response 1: We added more information (see the caption) for Figure 1 as below:
Line 57~58: The region of C has been reported as Mudeungsan tuff in Lim et al. (2015).
Point 2: Lines 59-62 Please, explain the abbreviations in the caption of Figure 2;
Response 2: We explained like below
Figure 2. Photographs of sampling locations at Mudeungsan A) Ipseokdae rock (ISD) and Jiwang peak (Jiwang). Talus was formed by the collapse of columnar joints. Ice Rock (IR) is located over this photograph. B) Baekma ridge (BN), Nakta peak (NT), and Anyang Mt. (Anyang). BN is the ridge of the mountain. C) Close-up view of ISD. D) View of IR. This was observed along the trail.
Point 3: Line 63 Methodology – synchrotron XRD method should be extended of description of the analytical errors particularly for substances of quantity below 1% , R parameter should be explain.
Response 3: Lines 72-75, 79-83, 96-103, Paragraph is accordingly extended to explain why very weak signal of substances can be detected, what is R parameter, and how analytical errors can be calculated.
Point 4: Line 64 How many samples have been analyzed?
Response 4: 9 samples were analyzed.
Point 5: Line 109 there is not possibly to show altered minerals without microscopic studies. Zeolites could be the primary one, as hydrothermal phases , the chlorites commonly forms from biotite thus biotite is altered to chlorite
Response 5: Zeolite is very well-known secondary minerals by hydrothermal alteration in volcanic rocks or ash layers.
Point 6: Line 112 in the Table 1 the contents of sanidine is completely different
Response 6: Line 125, It is description about quantitative results of Table 2. We corrected typos about contents of sanidine in this sentence.
Point 7: Line 115 not clear which minerals has suffered alteration, mineralogical description and petrography of the studied here samples is lacking.
Response 7: Line 127-131, We accordingly describe which minerals can suffer alteration in detail.
Point 8: Line 120. Low cristobalite can be pedogenic origin, it is observed as a detrital mineral in soils formed on the tuffs and occurs commonly as detrital grains in soils that formed on pyroclastic volcanic materials (Cronin et al., 1996, Sommer et al., 2006, Gutierrez – Castorena, Effland 2010) and this possibility should be taken to account into discussion as the sample were taken from the surface.
Response 8: Samples were taken from not surface or soil but from columnar joint itself. We therefore expected low cristobalite was primarily formed from volcanic process, not pedogenic origin.
Point 9: Line 128 synchotron X-Ray diffraction instead of XRD
Response 9: We accordingly changed.
Point 10: Lines 147 - 150 what about devitrification in studied tuffs ?
Response 10: This means rapid cooling of minerals from a glassy state, commonly reported in explosive volcanic eruptions.
Point 11: Line 154 – the explosive eruption have not been proved in this work.
Response 11: The very existence of tuff is evidence of an explosive eruption. This is because lava flows appear in an eruption type eruption.
Reviewer 2 Report
The communication “Mineralogy of the Mudeungsan tuff (Republic of Korea) using synchrotron X-ray diffraction and Rietveld quantitative phase analysis” presents an XRF and XRD characterization of Mudeungsan tuff. It is a short communication, which presents for the first time a characterization of these materials with these techniques. However, I consider that the data presented by XRD are not consistent with those presented by XRF and the work needs a major revision in order to be published. Analysis using other techniques could help to better understand the results obtained and a more adequate characterization of the phases present.
Coments:
Line 90: An R-factor less than 10% is specified for all samples, but as shown in Table 2 and Figure S1b for the ISD-2 sample it is 11.7. That should be clarified in the text. In addition, for the Jiwang sample, according to Table 2 and Figure S1h, the value is 8, with the ISD-2 sample being the two with the worst results, with the rest of the samples presenting a good figure of merit. A discussion in the text that explains why these two samples are different from the rest should be carried out.
The results presented by XRF in Table 1 are very homogeneous for all samples, which is discussed in the text. The results presented qualitatively in Figure 4 for XRD are also very homogeneous, except for small differences in phases such as Biotite, which appears with a clear preferential orientation in the ISD-2 sample, which is common in the 001 reflections of the phyllosilicates, or small differences in some other phases in low proportion. In general it can be said that those qualitative results presented in Figure 4 for XRD are consistent with those presented for XRF in Table 1. This has not been discussed in the text, and I think it could be incorporated, including the comment on the possible Biotite's preferential orientation in the ISD-2 sample.
However, the results presented in Table 2 by quantification by XRD seem completely inconsistent with those presented in Table 1 by XRF and those presented in Figure 4. Table 2 does not present homogeneous results in the quantification of the phases, which as seen in Table 2 and Figure 4, should be homogeneous if the samples analyzed were homogeneous. A coherent discussion should be carried out in the text to explain these differences.
In addition, other observations can be made on the results presented in Figures S. In Figures S1a, S1d, S1e and S1f, the samples present a clearly higher background noise than the rest of the samples. Has any treatment been carried out on this background in the rest of the samples? Are the differences due to the presence of more amorphous material in the samples presented in Figures S1a, S1d, S1e and S1f? I think that observations could be discussed in the text. Furthermore, no comment or measurement is presented in the text on the amount of amorphous material present in the samples, which could explain differences between the results obtained by XRF and XRD.
In the data shown in Table 2, the Jiwang sample presents very low amounts of Quartz compared to the rest of the samples, as well as a very high amount of Plagioclase and a very low amount of Sanidine. It is clearly the case furthest from the mean, but clear differences are also observed in these phases in the ISD-1, IR-1, BN-1 and BN-2 samples with respect to the others. What are these big differences due to? Why does the Jiwang sample show such a high percentage of Biotite? A discussion in the text on all these questions and a comparison with the results presented by XRF in Table 1 and also by XRD in Figure 4 would be important for the text to be more consistent.
Author Response
Point 1: Line 90: An R-factor less than 10% is specified for all samples, but as shown in Table 2 and Figure S1b for the ISD-2 sample it is 11.7. That should be clarified in the text. In addition, for the Jiwang sample, according to Table 2 and Figure S1h, the value is 8, with the ISD-2 sample being the two with the worst results, with the rest of the samples presenting a good figure of merit. A discussion in the text that explains why these two samples are different from the rest should be carried out.
Response 1: line 96-106, R-factors of ISD-2 and Jiwang samples were comparatively higher due to unidentified peaks. We described in experimental section with explanation of R-factor.
Point 2: The results presented by XRF in Table 1 are very homogeneous for all samples, which is discussed in the text. The results presented qualitatively in Figure 4 for XRD are also very homogeneous, except for small differences in phases such as Biotite, which appears with a clear preferential orientation in the ISD-2 sample, which is common in the 001 reflections of the phyllosilicates, or small differences in some other phases in low proportion. In general it can be said that those qualitative results presented in Figure 4 for XRD are consistent with those presented for XRF in Table 1. This has not been discussed in the text, and I think it could be incorporated, including the comment on the possible Biotite's preferential orientation in the ISD-2 sample.
Response 2: line 126-129, We described preferred orientation of (001) reflection of biotite in ISD-2 samples, and how to avoid miscalculation of quantification.
Point 3: However, the results presented in Table 2 by quantification by XRD seem completely inconsistent with those presented in Table 1 by XRF and those presented in Figure 4. Table 2 does not present homogeneous results in the quantification of the phases, which as seen in Table 2 and Figure 4, should be homogeneous if the samples analyzed were homogeneous. A coherent discussion should be carried out in the text to explain these differences.
Response 3: line 122-124, 131-140, significant difference is observed at sanidine and biotite in Jiwang sample. In figure 4, intensity of peaks of biotite(B) and sanidine(S) are stronger (at ca. 0.6Å-1) and weaker (at ca. 1.55Å-1 and ca. 1.82Å-1…) than in other samples, respectively. The Jiwang sample is slightly more mafic than other samples, containing more plagioclase. For this reason, the Mudeungsan tuff is mostly dacitic, but Jiwang might be not. We modified the sentences simpler.
Point 4: In addition, other observations can be made on the results presented in Figures S. In Figures S1a, S1d, S1e and S1f, the samples present a clearly higher background noise than the rest of the samples. Has any treatment been carried out on this background in the rest of the samples? Are the differences due to the presence of more amorphous material in the samples presented in Figures S1a, S1d, S1e and S1f? I think that observations could be discussed in the text. Furthermore, no comment or measurement is presented in the text on the amount of amorphous material present in the samples, which could explain differences between the results obtained by XRF and XRD.
Response 4: line 139-140, In Figures S1a, S1d, S1e and S1f, it is just due to different y-axis scale when we plotted graphical results. We also thought that amorphous materials possibly coexist in all sample. We plan to carry out quantification of amorphous materials in our further study.
Point 5: In the data shown in Table 2, the Jiwang sample presents very low amounts of Quartz compared to the rest of the samples, as well as a very high amount of Plagioclase and a very low amount of Sanidine. It is clearly the case furthest from the mean, but clear differences are also observed in these phases in the ISD-1, IR-1, BN-1 and BN-2 samples with respect to the others. What are these big differences due to? Why does the Jiwang sample show such a high percentage of Biotite? A discussion in the text on all these questions and a comparison with the results presented by XRF in Table 1 and also by XRD in Figure 4 would be important for the text to be more consistent.
Response 5: line 133-141, SIO2 contents in XRF is very low in Jiwang. It may be reflected to low quantification of quartz. In combined results both the XRF (Table 1) and quantification results (Table 2) about the Jiwang sample, we expect two possibilities. First one is that a little portion of andesite may coexist in Jiwang tuff, and therefore plagioclase and biotite is more abundant than alkali feldspar such as sanidine. So, quantification result of Jiwang is different from other XRD results whereas composition of CaO, Na2O, and K2O in XRF result is similar. Second possibility is that lack of time for crystallization of sanidine at Jiwang tuff. It can be reflected to amorphous hump in diffraction data. Further studies such as calculation of amorphous phase and XRF using thin section is ongoing to understand in detail.
Round 2
Reviewer 1 Report
The manuscript can be published in present form.
Author Response
Q. The manuscript can be published in present form.
A. Thank you very much for you kindness!
Reviewer 2 Report
The work is currently a short communication and I think could be improved and provided with a more extensive work.
Although a review of the work has been carried out, I consider that it has not been possible to improve the main aspects that I observed in the first review. The results of quantification of the phases by XRD presented, especially for the Ji-Wang sample, do not seem consistent to me. I think it would be of interest to include the analysis of the amount of amorphous to obtain a more correct work on the interpretations made. In addition, the use of other techniques could help in the interpretation of the results. Since the work has been done on a high intensity synchrotron source, the authors could try to identify the phases present even to a lesser proportion. The results obtained do not present great advantages to those obtained in conventional diffraction.
Author Response
Q. The work is currently a short communication and I think could be improved and provided with a more extensive work.
A. Thanks for your comments. As we submitted our manuscript to “Communication of Applied Sciences”, we should have made our stuff short and concise. I believe you understand this. Like you suggested, we are preparing another paper with petrologic and isotopic results. If you have a chance, you will see our extensive work there.
Q. Although a review of the work has been carried out, I consider that it has not been possible to improve the main aspects that I observed in the first review. The results of quantification of the phases by XRD presented, especially for the Ji-Wang sample, do not seem consistent to me. I think it would be of interest to include the analysis of the amount of amorphous to obtain a more correct work on the interpretations made. In addition, the use of other techniques could help in the interpretation of the results. Since the work has been done on a high intensity synchrotron source, the authors could try to identify the phases present even to a lesser proportion. The results obtained do not present great advantages to those obtained in conventional diffraction.
A. Now, I figured it out. I have checked the thin section of Jiwang. Then, I recognized there are few biotite grains in the samples. I agree with your point on this issue. I and my colleagues decided not to take Jiwang anymore in this manuscript. Please check that I excluded the Jiwang part. I really appreciate you for pointing out that this manuscript can no longer have this erroneous result.
Round 3
Reviewer 2 Report
The authors have again reviewed the work and I consider that the interpretation of XRD presented is more appropriate by removing the sample that had more interpretation errors. I see the rest of the work well, being a short communication as the authors explained.